# L-Citrulline Supplementation and Exercise in the Management of Sarcopenia

**DOI:** 10.3390/nu13093133

**Published:** 2021-09-08

**Authors:** Alberto Caballero-García, Jorge Pascual-Fernández, David César Noriega-González, Hugo J. Bello, Antoni Pons-Biescas, Enrique Roche, Alfredo Córdova-Martínez

**Affiliations:** 1Department of Anatomy and Radiology, Faculty of Health Sciences, GIR: “Physical Exercise and Aging”, Campus Universitario “Los Pajaritos”, University of Valladolid, 42004 Soria, Spain; alberto.caballero@uva.es; 2Medical Extrahospital Emergency Service of Navarra, 31500 Pamplona, Spain; jpfoliva@gmail.com; 3Department of Surgery, Ophthalmology, Otorhinolaryngology and Physiotherapy, Faculty of Medicine, Hospital Clínico Universitario de Valladolid, 47003 Valladolid, Spain; davidcesar.noriega@uva.es; 4Department of Mathematics, School of Forestry, Agricultural and Bioenergy Engineering, GIR: “Physical Exercise and Aging”, Campus Universitario “Los Pajaritos”, University of Valladolid, 42004 Soria, Spain; hjbello.wk@gmail.com; 5Research Group on Community Nutrition and Oxidative Stress, University of Balearic Islands, 07122 Palma de Mallorca, Spain; antonipons@uib.es; 6CIBER Fisiopatología de la Obesidad y Nutrición (CIBEROBN), Instituto de Salud Carlos III, 28029 Madrid, Spain; 7Department of Applied Biology-Nutrition, Institute of Bioengineering, Miguel Hernández University, 03202 Elche, Spain; 8Alicante Institute for Health and Biomedical Research (ISABIAL Foundation), 03010 Alicante, Spain; 9Department of Biochemistry, Molecular Biology and Physiology, Health Sciences Faculty, GIR: “Physical Exercise and Aging”, Campus Universitario “Los Pajaritos”, University of Valladolid, 42004 Soria, Spain

**Keywords:** aging, muscle wasting, physical activity, sarcopenia

## Abstract

Sarcopenia is a process associated to aging. Persistent inflammation and oxidative stress in muscle favour muscle wasting and decreased ability to perform physical activity. Controlled exercise can optimize blood flux and moderate the production of reactive oxygen species. Therefore, supplements that can work as a vasodilators and control oxidative stress, might be beneficial for active elders. In this context, we have tested citrulline supplementation in a group of 44 participants aged from 60–73 years that followed a physical activity program adapted to their age and capacities. Volunteers were divided in two groups: placebo (*n* = 22) and citrullline supplemented (*n* = 22). Different physical tests and blood extractions were performed at the beginning and at the end of intervention (six weeks). Strength and endurance showed a tendency to increase in the citrulline supplemented group, with no significant differences respect to placebo. However, walking speed in the citrulline supplemented group improved significantly compared to placebo. Markers of muscle damage as well as circulating levels of testosterone, cortisol and vitamin D showed no significant changes, but a tendency to improve at the end of intervention in the supplemented group compared to placebo. Additional studies are necessary to confirm the effect of citrulline supplementation in sarcopenia delay.

## 1. Introduction

Sarcopenia is a process associated to aging that starts from the age of 30 years and progresses resulting in strength reduction due to a decrease in muscle mass [1]. Muscle wasting and degeneration is a consequence of muscle atrophy and muscle cell death, leading to the loss of strength and muscle mass [2,3,4]. The European Working Group on Sarcopenia in Older People (EWGSOP) proposed the use of an algorithm associated with loss of walking speed and grip strength to define the effect of sarcopenia [2,5]. In addition, sarcopenia is multifactorial and involves age, genetics, nutrition, hormonal changes, inflammatory processes, insulin resistance, physical abilities, lifestyle, etc. [3,6,7].

From the molecular point of view, the mitochondrial electron transport chain is a key element in ATP production and a determining factor in the ageing process. Malfunctioning of this chain leads to a suboptimal cellular energy production, explaining the increased difficulty of affected people in performing motor activities [8]. In addition, the production of pro-inflammatory cytokines generates a persistent inflammatory state that leads to muscle wasting and the development of sarcopenia [9]. The increased production of cytokines, particularly interleukin-6 (IL-6) and tumour necrosis factor-α (TNF-α), has been associated with an oxidative stress state [10,11]. The control of reactive oxygen species (ROS) production by the intracellular antioxidant systems decreases with aging, favouring the development of inflammation and activation of catabolic processes typical of sarcopenia [12]. In addition, RNS (reactive nitrogen species) are also produced through the activity of nitric oxide synthase (NOS). The enzyme takes arginine (Arg) as substrate and produces citrulline and nitric oxide (NO), a potent vasodilator [13,14,15].

ROS and RNS increase during exercise [16,17]. In this context, a moderate production is necessary to favour an adaptive response of the muscle antioxidant systems for subsequent exercise routines. However, an uncontrolled ROS/RNS production would affect muscle contraction reducing strength capacity. Therefore, training exercises need to be planned to reach a fine balance between ROS and RNS generated in muscle contraction, exerting a key modulation on muscle function [12,18].

Citrulline is a non-essential amino acid metabolised by the kidney and giving rise to Arg, a key metabolite in NO synthesis and urea cycle [19,20,21,22,23,24,25]. Arg metabolised in the liver stays for urea synthesis and for protein synthesis in the rest of tissues. Citrulline from kidney provides new Arg for NO production and vasodilation processes [26,27]. Therefore, citrulline could be considered as an Arg precursor. In this process, glutamine plays a key role closing citrulline metabolism in the urea cycle [28]. The amount of citrulline converted in the kidney is sufficient to cover the needs of the whole organism by constantly recycling intestinal citrulline and secondarily maintaining the concentration of Arg for NO production [28].

In this context, the release of pro-inflammatory cytokines in situations of cellular stress, leads to reduced tissue regeneration and failures in cellular metabolism. Citrulline reduces serum concentrations of IL-6, TNF-α and C-reactive protein (CRP), which increase with age and physical exercise [29]. Citrulline supplementation induces vascular protection through NO production by suppressing endothelial damage [30]. Citrulline exerts also an antioxidant action, favours protein synthesis, has an anti-inflammatory effect and promotes aerobic metabolism. After the administration of citrulline, an increase in oxygenation at muscle level (through the production of NO) has been observed [31,32,33].

Moreover, in the field of sport, creatinine kinase (CK), as well as other proteins such as alanine aminotransferase (ALT), aspartate aminotransferase (AST), lactate dehydrogenase (LDH) and myoglobin, are considered markers of muscle damage [34]. Increases in these enzymes may represent an index of necrosis and tissue damage following acute as well as chronic muscle injuries [35]. Therefore, considering these muscle markers as indicators of muscle status, they are very useful to follow the evolution of sarcopenia just using the regular blood analysis performed usually in laboratories and health centres [36].

All of this evidence might suggest a possible role of citrulline as a candidate supplement to control sarcopenia. We addressed this question in the present report by studying a group of healthy elders over 60 years of age supplemented with citrulline and performing a physical activity protocol suitable for their age range. To this end, we determined different parameters related to muscle integrity and inflammation.

## 2. Materials and Methods

### 2.1. Participants

The study involved 44 subjects (26 women and 18 men) with an age range of 60–73 years old. Anthropometric parameters and blood pressure values at the beginning of the intervention are shown in Table 1. All participants were regularly engaged in a physical activity programme of the Soria City Council. Before intervention, a meeting was held to explain the study, objectives, methodology and commitment. At the end of the meeting, the addresses of 2 health centres were provided for recruitment of interested people. Those who agreed to voluntarily participate underwent a medical check-up, which was then contrasted and supervised with their family doctor. Inclusion and exclusion from the study was stablished according to the inclusion and exclusion criteria and in accordance with the family doctor. Before intervention, participants were distributed in 2 groups (*n* = 22 per group): placebo group (PL) (1 man and 21 women) and citrulline-malate supplemented group (CM) (17 men and 5 women). This unequal distribution of genders occurs because the intervention was performed in 2 different health centres with no possibility to move volunteers from one place to the other. This could be considered as a limitation of the study. All volunteers were committed and none of them dropped out from the study.

The study was approved by the Clinical Research Ethics Committee (CEIC) of the University of León (Ref: ETICA-ULE-021-2020) and all subjects signed a written informed consent. The inclusion criteria were persons over 60 years of age and in healthy state (not affected by any exclusion criteria). Exclusion criteria were as follows: (a) story of dementia (suspected by MAP setting and diagnosed); (b) moderate/severe chronic obstructive pulmonary disease (COPD) with Bodex index C or D; (c) functional limitation by the Barthel scale (less than 100 = maximum score) and Lawton–Brody scale (less than 8 = maximum value); (d) recent acute myocardial infarction (3–6 months) or unstable angina; (e) uncontrolled atrial or ventricular arrhythmias, dissecting aortic aneurysm, severe aortic stenosis, acute endocarditis/pericarditis; (f) uncontrolled hypertension (>180/100 mm Hg); (g) acute thrombo-embolic disease; (h) acute/chronic heart failure with NYHA (New York Heart Association) scale >II; (i) acute/chronic respiratory failure; (j) uncontrolled orthostatic hypotension; (k) diabetes mellitus with acute decompensation or uncontrolled hypoglycaemia; (l) recent bone fracture (last month); (m) previous supplementation with amino acids or other nutritional compounds to improve physical performance; (n) any other circumstance deemed by their physician to preclude physical activity.

The study was a double-blind, placebo-controlled trial. Citrulline malate (3 g/day) or placebo was administered orally for 6 consecutive weeks. Blood controls and physical assessment tests were performed at the beginning of the study and at the end of the 6th week of the intervention.

For physical activity monitoring, a record was made at each of the training sessions. One of the members of the research team participated and collected the required information at the end of each session. In this way, we assessed the type and intensity of the exercises and the subjective sensation of each participant. The distribution of times and workloads is shown in Table 2.

### 2.2. CM Supplementation

Supplementation was carried out with CM (3 g/day) in capsules. The placebo was prepared with lactose and starch in capsules of the same size, weight and colour than CM capsules. Each week a member of the research team (physician) was in charge of the weekly distribution of capsules and the recording of any clinical observations that might occur. The doctor in charge of distribution was unaware of the composition of the capsules during the entire intervention period. Capsules were blinded by the codes A or B. CM group (*n* = 22) and PL group (*n*= 22) took the corresponding capsules during the 6-week follow-up period.

### 2.3. Physical Tests

Assessment of physical abilities was performed as follows: (a) 6-min test (endurance) was performed on an approved 400-m athletics track (Spanish Athletics Federation) and the distance covered after 6 min was measured; (b) hand grip strength by JAMAR digital dynamometer (0–90 Kg); (c) gait speed, a distance of 4 m was measured by installing photoelectric cells at both ends. To minimise variability, volunteers were asked to start walking 5 m before the timed area at their usual walking speed; (d) squat test that determines the time taken to perform 5 full squats from a seated position, without allowing arm assistance, at the fastest possible speed; (e) short physical performance battery (SPPB) frailty test, which in turn consists of 3 tests measuring balance in three different positions, walking speed and leg strength. The sum of the obtained scores determines the level of frailty in the subject; (f) balance test that determines the ability to maintain specific positions (standing, semi-tandem and tandem position) for at least 10 s. All individuals used the same model of chair with a height of 45 cm clearance to the floor.

### 2.4. Blood Parameters

The first blood extraction was performed the first day that participants started the study and the second extraction was performed the last day of the intervention, after 6 weeks. The blood samples were taken at 08:00 AM with participants in fasting conditions and with an adequate rest the previous night of 8–12 h. Serum (10 mL) was obtained in vacutainer tubes containing clot activator. Additionally, 3–5 mL of blood was placed in EDTA tubes to obtain plasma. Blood extractions were performed while the participants remained seated. Plasma tubes were stored in refrigerated containers at a temperature of 4 °C and serum tubes were frozen at −20 °C. Haematological analysis was performed on a System Coulter Counter MAX-M counter.

Circulating proteins include ALT, AST, CK and LDH as indicators of muscle damage, and total proteins (TP) as indicators of immune function as well as liver and kidney status. Circulating metabolites include glucose, urea, uric acid, total cholesterol, LDL-cholesterol and HDL-cholesterol. All were determined using the Architect ci8200^®^ analyser. Testosterone, cortisol and vitamin D were analysed in an Architect 2000^®^ system. Ferritin was analysed using the IRMA commercial kit (Bio Rad^®^, Madrid, Spain) using duplicate serum aliquots. For serum iron, the Syncrhon Cx analyser (Beckman^®^, Brea, CA, USA) was used. Anthropometric parameters were recorded 15 min after blood collection.

### 2.5. Statistics

Statistical analyses were performed using IBM Statistical Package (SPSS Version 24). Data were expressed as mean ± standard error of the mean (X ± SEM). After checking for normal distribution using the Shapiro–Wilk and Kolmogorov–Smirnov test, Levene’s homogeneity of variances was studied. Frequencies were estimated for categorical variables and averages of central tendency and standard deviation were calculated for quantitative variables. The effect of citrulline supplementation on changes in muscle tissue was tested by two-way ANOVA test with supplementation (PL vs. CM) and 6 weeks of training (beginning vs. end of intervention) as factors, performing as well a post hoc test (Bonferroni and Scheffé) on those variables that showed a level of statistical significance. In all calculations the level of statistical significance was *p* < 0.05.

## 3. Results

The physical tests analysed in the present study encompass different parameters that attempt to monitor together the different capacities commonly used for the study of physical performance and sarcopenia. We assessed strength, endurance and speed (Table 3).

Overall strength and endurance increased slightly in the CM group, showing a tendency. Nevertheless, differences were not significant comparing both groups (PL vs. CM) or comparing the beginning vs. the end of the study. However, walking speed in the CM group was significantly lower. Regarding frailty (SPPB), which ultimately refers to the functional capacity of the elderly, we found no significant differences between the two groups and comparing beginning vs. end of the study (Table 3). The same results were obtained for the squat test (Table 3). The balance test at the beginning of the study indicated correct scores for all participants that did not change at the end of the study (not shown).

Regarding haematological parameters (Table 4), no significant differences were observed comparing the beginning vs. the end of the study into the same group as well as comparing PL vs. CM at the end of intervention. Only a modest tendency to increase in haematocrit and haemoglobin levels was noticed in the CM group compared to PL, but differences were not significant. No significant differences were observed as well in serum iron and ferritin, indicating that participants had no anaemia at the beginning of the study or developed anaemia during the intervention. The rest of circulating variables showed no significant differences (not shown).

Figure 1 shows the results obtained for the circulating markers of muscle damage. We did not observe significant variations between the groups (PL vs. CM). In addition, no significant changes were noticed when comparing the beginning vs. the end of the intervention (not shown). Moreover, the values found were into the healthy range. This is probably due to the fact that the exercise protocol was not highly demanding, avoiding muscle damage and preserving its function. In addition, participants started from a good baseline physical condition.

Figure 1 shows no significant changes in the muscle enzymes ALT, AST and CK, as well as in TP. Nevertheless, while remaining at physiological levels, we observed slight increases in CK and ALT. It is possible that the slight increase in CK in the CM group is related to an increase in work intensity as a consequence of the improvement in the perception of effort.

Figure 2 shows the results for the circulating levels of the hormones testosterone, cortisol, and vitamin D at the end of intervention. Lower but not significant, levels of cortisol were observed in CM compared to PL, suggesting a tendency to decrease in the stress associated to exercise. In addition, higher but not significant levels of testosterone were observed in CM compared to PL, suggesting a tendency of the CM group to assimilate optimally the programmed exercise intensity. This is confirmed by the testosterone–cortisol index indicating a prevalence of protein anabolism over protein catabolism. Regarding vitamin D (normal circulating levels are 20–40 ng/mL), slightly higher values were observed in the CM group, suggesting a tendency for an optimal protein turnover in muscle and a better post-exercise recovery [37,38]. The proposed mechanism seems to indicate that sarcopenia is related to a decrease in vitamin D receptor in muscle, leading to a reduction in post-exercise recovery [37,38]. No significant changes were observed as well comparing the values at the beginning vs. the end of intervention into each group (not shown).

## 4. Discussion

The most relevant observation of the present study was that the volunteers of the CM group displayed a tendency to improve in their physical performance compared to the PL group. In particular, the walking speed test showed significant differences. The association of CM supplementation (3 g/day) for six weeks, associated with a programmed protocol of physical activity adapted to elders, resulted in a general tendency to improve in terms of adaptation to exercise and optimal recovery.

Several authors, using higher doses of CM (8 g/day) showed a clear improvement in strength [39,40]. However, these data were not reproduced in a study with a similar design carried out on women [41]. This disparity leads us to hypothesise that lowering the dose to 6 g/day may be the reason why Cutrufello et al. [41] did not find the performance benefits cited in other studies with male volunteers using higher doses (8 g/day). We did not observe significant increases either, only slight increases (a tendency), but the dose used was the lower (3 g/day) compared to other studies [39,40,41]. Glen et al. [42] carried out a study only with young (23 ± 3 years) trained female volunteers, who were supplemented with 8 g/day of CM and performed a submaximal bench press test until exhaustion. These authors observed similar results to those obtained in groups of men. This leads us to believe that doses close to 0.12 g/kg are adequate for the improvement of strength-endurance in trained individuals. However, Suzuki et al. [43], also in young people (18–25 years) showed significant results with very small doses (1.2 g/day) maintaining supplementation for seven days. However, in this study, CM was combined with 1.2 g/day of Arg. The authors propose that an optimal blood NO concentration can be achieved with the combination of both amino acids, leading to an improvement in performance.

In our study, carried out in older people (60–73 years), with more women than men, we used for both genders about 0.04 g of citrulline/kg only for six weeks. Otherwise said, we used much lower doses than those used in the studies cited before. Therefore, it is predictable obtaining less significant increases. Moreover, 3 g of citrulline/day did not reach a sufficient cumulative effect over the six weeks, which somewhat limits the observation of a significant improvement. However, from a psychological point of view, the increases found are significant, especially in terms of the sensations expressed by participants of the CM group (not shown).

The effects of CM have been observed mainly in young individuals. In this context, it is important to mention that NO production decreases with age [44], due to increased arginase activity [45]. Therefore, citrulline supplementation could be expected to increase Arg concentration, which may potentially increase NO production from NOS activity, improving thereby physical fitness [44,45]. We believe that in our study this phenomenon might be a limiting factor in improving the physical condition of the studied subjects.

In addition, the older people participating in our study performed a maintenance exercise protocol (around 60% of their maximum capacity). This design can explain why the results were not significant compared to designs of greater intensity and work volume usually programmed for young people.

Regarding circulating parameters, neutrophil/lymphocyte ratio does not indicate an inflammatory component [46]. In addition, the circulating muscle biomarkers did not display significant changes in any of the enzymes analysed. Although, CK levels increased in the group treated with CM compared to PL, but levels were into healthy ranges.

Although CM stimulates NO production and improves exercise performance by reducing muscle damage rates [44,45], the direct benefits on antioxidant and muscle markers are unclear. Chronic CM supplementation has been shown to improve skeletal muscle strength concomitantly with increased oxidative energy turnover, lower pH-to-power ratio [47] and lower ATP cost [48]. These data suggest that CM supplementation may improve skeletal muscle metabolism and/or contractile efficiency, which would be expected to predispose to an optimal adaptation to fatigue [49]. In our study, we observed a modulating effect on markers of muscle damage accompanied by an improvement in physical fitness in subjects over 60 years of age. These results are coming together with a subjective sensation of reduced muscle soreness and increased exercise tolerance in CM participants (not shown).

The effect of a slightly increase in the circulating levels of CK, AST and ALT in the group supplemented with CM compared to PL does not pose any difficulty, because the tendency of improvement of CM group in the physical tests was higher than in the PL group. We hypothesize that this may be due to the fact that in the CM group, the participants carried out the exercise program with more involvement and more intensity, leading to an improvement in sensations. Despite being a double-blind study, minimal positive feelings lead to greater stimulation for the development of the physical activity program.

Regarding the hormonal response, we observed that testosterone and vitamin D increased along with a decrease in cortisol levels. Cortisol levels are considered to be indicative of stress intensity [50]. Testosterone is an index of body regeneration [51,52,53]. The testosterone/cortisol ratio is an indicator of anabolic/catabolic balance, and has also been proposed as an indicator of adaptation to training [54]. In sarcopenia, there is a loss of muscle mass and strength, and testosterone could play a key role in the maintenance of both [55], which progressively decrease with age [56,57].

In our study, although the results showed no significant variations, there was a tendency for predominance of anabolic over catabolic effects. We observed a tendency of testosterone to increase in the CM group, accompanied by a decrease in cortisol (lower stress). In addition, the testosterone/cortisol ratio showed a predominance of anabolic effects (4.41 in CM vs. 2.39 in PL) (Figure 2). These results have clinical relevance both from an objective point of view (improvement in physical condition) and subjectively in terms of the patient’s perception of adaptation to exercise. We believe that with a larger recruitment of volunteers and a longer follow-up period, these differences could become significant.

Low vitamin D levels are associated with decreased muscle mass, decreased muscle strength, loss of balance control and therefore increased risk of falls [58,59]. There are references linking low serum 25-hydroxy vitamin D concentration and sarcopenia [58]. Vitamin D deficiency may decrease muscle protein anabolism [6]. In our study, it was relevant that vitamin D concentration presented a tendency to increase in the CM group. This could be related to the anabolic effect of vitamin D and a possible accumulation in muscle due to the effect of citrulline. Altogether, these results might account for the physical and psychological improvement observed in the CM group.

In conclusion, and in view of the results obtained, we can deduce that well-controlled and adapted physical activity is the most effective instrument to delay the decline of the physical capacities associated to sarcopenia progression: strength and endurance. Supplementation with amino acids such as citrulline (and likely Arg) and/or changes in nutritional habits can improve the effects of exercise. Optimal changes are reflected in a better anabolic situation that can help in post-exercise recovery on this population segment.

## 5. Conclusions

Taking into account the characteristics of our study, the results obtained show a modest potency. Nevertheless, in this line we also found studies which demonstrate a significant benefit in the use of amino acids such as citrulline [40,47], and yet in other studies no statistically significant benefit is achieved [60]. We believe that this disparity may be due to differences in the target population, exercise program, doses used for supplementation and time of intervention, among others. Additional research is necessary to answer all these questions. Beneficial effects of citrulline in combination with exercise in the management of sarcopenia remain to be determined.

## Figures and Tables

**Figure 1 nutrients-13-03133-f001:**
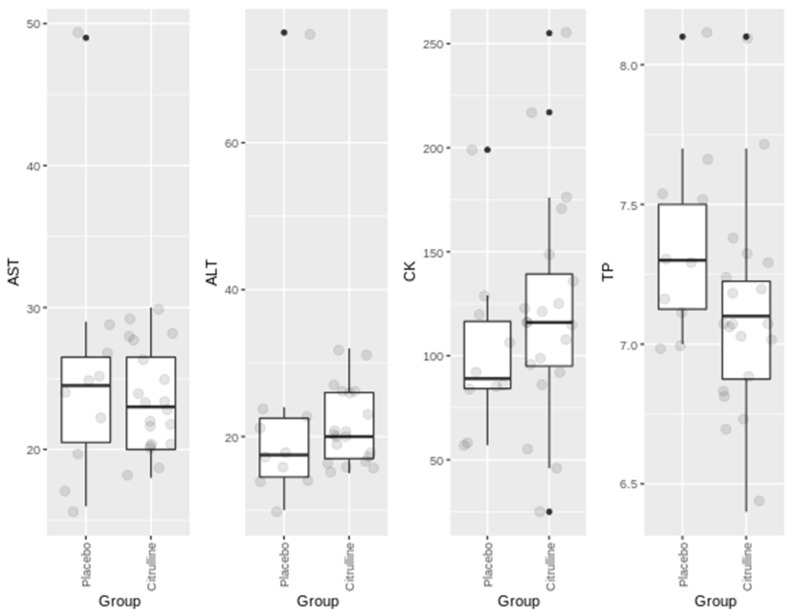
Boxplots of circulating muscle markers at the end of intervention in the placebo and in the citrulline supplemented group. AST (alanine aminotransferase) and ALT (aspartate aminotransferase) are expressed in IU/L. CK (creatine kinase) is expressed in m IU/L. TP (total proteins) are expressed in g/dL.

**Figure 2 nutrients-13-03133-f002:**
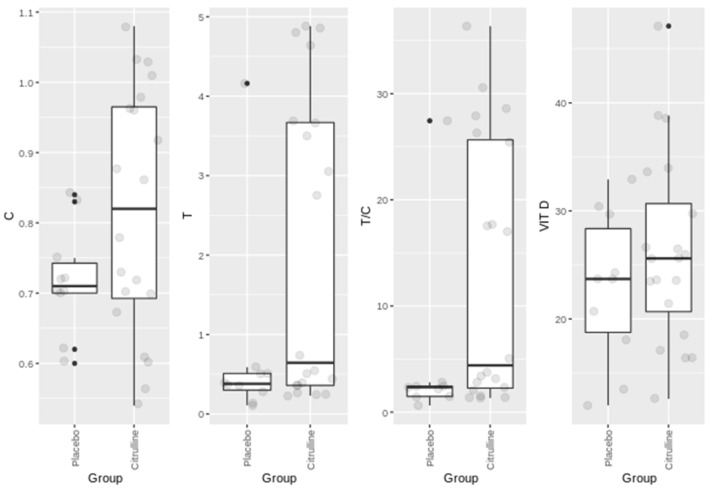
Boxplots of circulating levels of cortisol (C) expressed in ng/mL, testosterone (T) expressed in ng/mL, and vitamin D (Vit D) expressed in ng/mL in the placebo group and in the citrulline supplemented group at the end of intervention.

**Table 1 nutrients-13-03133-t001:** Anthropometric parameters and blood pressure values of participants at the beginning of the study.

Parameter	Men	Women
*n*	18	26
Height (cm)	171.0 ± 4.3	158.1 ± 6.1
Age (years)	64.8 ± 3.6	65.4 ± 4.4
Weight (kg)	78.4 ± 8.9	63.9 ± 7.9
BMI (kg/m^2^)	26.6 ± 2.3	24.9 ± 2.9
% Fat	24.4 ± 2.7	34.9 ± 5.6
Systolic BP	133 ± 17	130 ± 21
Diastolic BP	79 ± 10	77 ± 11

Abbreviations used: BMI, body mass index; BP, blood pressure.

**Table 2 nutrients-13-03133-t002:** Physical activity protocol performed in each training session.

Session Protocol	Time (min)	Content	Level of Effort
Warm-up	10	General mobility, light movements	4
Balance	5	Standing and monopodial exercises	3
Aerobic endurance	10	Walking, slow running	7
Aerobic resistance	20	Overload exercises, with balls, dumbbells, elastic bands, steps	8

**Table 3 nutrients-13-03133-t003:** Physical condition tests of the studied participants at the beginning (*t* = 0) and at the end (6th week) of intervention in placebo and in citrulline supplemented groups.

Tests	PL	CM	
***t* = 0**			*** *p***
Endurance 6 min (m)	931.1 ± 181.5	925.8 ± 191.0	0.913
Strength-Dynamometer (Kg)	37 (29–46)	29.5 (26.5–37.5)	0.540
Speed (s)	2.3 (2.1–2.8)	2.1 (1.9–2.4)	0.365
Squat (s)	9.9 (9.6–11.4)	10.5 (10.0–11.8)	0.092
SPPB (score)	12 (11–12)	12 (11–12)	0.563
**6th week**			**** *p***
Endurance 6 min (m)	842.9 ± 208.9	969.8 ± 237.0	0.162
Strength-Dynamometer (Kg)	27.5 (25–29)	31 (25.5–46.5)	0.301
Speed (s)	2.4 (2.2–2.6)	2.1 (1.9–2.3)	0.038
Squat (s)	11.1 (4.5–12.8)	10.0 (8.8–11.3)	0.099
SPPB (score)	11 (11–12)	12 (11.5–12)	0.159

* *p* indicates significant differences at the beginning (*t* = 0) vs. the end (6th week) of the intervention. ** *p* indicates significant differences of PL vs. CM. Abbreviations used: CM, citrulline supplemented group; PL, placebo group; SPPB, short physical performance battery.

**Table 4 nutrients-13-03133-t004:** Haematological parameters and circulating iron and ferritin levels at the beginning (*t* = 0) and at the end (6th week) of intervention in placebo and in citrulline supplemented groups.

Parameter	PL	CM	
***t* = 0**			*** *p***
Hb (g/dL)	14.2 (13.4–15.2)	14.8 (14.35–15.45)	0.931
RBC (10^6^ cells/μL)	4.7 ± 0.3	4.8 ± 0.3	0.307
Hematocrit (%)	41.6 (40.2–44.5)	42.7 (41.5–44.9)	0.734
WBC (10^3^ cells/µL)	6.4 (5.7–8.0)	6.2 (5.2–6.8)	0.582
Lymphocytes (%)	28.7 (17.2–38.1)	34.3 (32.8–38.0)	0.254
Monocytes (%)	6.4 (4.7–9.4)	8.3 (7.3–9.1)	0.768
Neutrophils (%)	63.1 (50.5–71.1)	52.9 (49.1–55.0)	0.228
Serum iron (µg/dL)	78 (64–106)	89 (75–119.5)	0.347
Ferritin (ng/mL)	62 (40–113)	93 (81.5–193.5)	0.524
**6th week**			**** *p***
Hb (g/dL)	14.6 (14.2–15.0)	15.1 (14.4–15.5)	0.209
RBC (10^6^ cells/μL)	4.8 ± 0.2	4.9 ± 0.3	0.463
Hematocrit (%)	43.3 (41.8–44.2)	44.7 (41.6–45.5)	0.172
WBC (10^3^ cells/µL)	6.7 (5.5–7.2)	6.8 (5.6–7.9)	0.441
Lymphocytes (%)	34.4 (30.9–38.8)	36.2 (29.5–40.2)	0.441
Monocytes (%)	8.1 (7.7–9.6)	8.6 (7.1–9.2)	0.947
Neutrophils (%)	54.1 (50.1–57.9)	53.3 (50.1–56.5)	0.843
Serum iron (µg/dL)	96.5 (81–108)	93.5 (83–100)	0.774
Ferritin (ng/mL)	86.5 (47–120)	91 (48.5–181.5)	0.843

* *p* indicates significant differences at the beginning (*t* = 0) vs. the end (6th week) of the intervention. ** *p* indicates significant differences of PL vs. CM. Abbreviations used: CM, citrulline supplemented group; Hb, haemoglobin; PL, placebo group; RBC, red blood cells; WBC: white blood cells.

## Data Availability

The data that support the findings of this study are available from the corresponding author, upon reasonable request.

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
