# Peer review of "L-Citrulline Supplementation and Exercise in the Management of Sarcopenia"

_nutrients, 2021, doi:10.3390/nu13093133_

Round 1

Reviewer 1 Report

The present manuscript evaluates the effects of citrulline supplementation combined to exercise in the management of sarcopenia. While citrulline supplementation has showed some positive effect on muscle function and performance in trained (mostly young) people. Potential beneficial effects of citrulline supplementation on muscle function in elder people haven’t been explored yet and it therefore represents a novel and original line of research. Participants of the study are subjected to physical activity and citrulline supplementation and authors evaluate their physical performance, haematological parameters and some circulating factors representative of muscle function.

Results of the study are disappointing as only one of all parameters measured is statistically significant. Importantly, the absence of error bars on the two figures makes it very difficult to draw conclusions on the relevance of these data. I would have also appreciated more explanation on the choice of the dose of citrulline-malate (3g/day) administrated which is way inferior to doses that already showed beneficial effects in the literature (8g/day).

The conclusion that “controlled and adapted physical activity is the most effective instrument to delay the decline of physical activity in sarcopenia” (line 331) is not relevant as the study did not compared activity vs no activity in sarcopenia but only, activity alone vs activity + citrulline. Beneficial effects of citrulline in combination with exercise in the management of sarcopenia remain to be determined.

Author Response

Reviewer 1 (round 1)

The present manuscript evaluates the effects of citrulline supplementation combined to exercise in the management of sarcopenia. While citrulline supplementation has showed some positive effect on muscle function and performance in trained (mostly young) people. Potential beneficial effects of citrulline supplementation on muscle function in elder people haven’t been explored yet and it therefore represents a novel and original line of research. Participants of the study are subjected to physical activity and citrulline supplementation and authors evaluate their physical performance, haematological parameters and some circulating factors representative of muscle function.

ANSWER: We appreciate these comments. The Reviewer is right indicating that citrulline supplementation works optimally in young people. However, it remained to be explored in old people. The main conclusion is that we cannot translate the results obtained in young people to elders. Likely, a longer time of supplementation could be necessary. In any case, this manuscript opens the way for new research in the field.

Results of the study are disappointing as only one of all parameters measured is statistically significant. Importantly, the absence of error bars on the two figures makes it very difficult to draw conclusions on the relevance of these data. I would have also appreciated more explanation on the choice of the dose of citrulline-malate (3g/day) administrated which is way inferior to doses that already showed beneficial effects in the literature (8 g/day).

ANSWER: We have remade Figures 1 and 2 as boxplots to incorporate Reviewer comments. Regarding the dose used, we have found several discrepancies in the literature as we have mentioned at the beginning of the discussion (Paragraphs 1-3). Since studies in elders are not available, we preferred to be cautious. In any case, we indicated as well at the beginning of Discussion that the dose could be a point to take into account for future research.

The conclusion that “controlled and adapted physical activity is the most effective instrument to delay the decline of physical activity in sarcopenia” (line 331) is not relevant as the study did not compared activity vs no activity in sarcopenia but only, activity alone vs activity + citrulline. Beneficial effects of citrulline in combination with exercise in the management of sarcopenia remain to be determined.

ANSWER: Reviewer is right. The sentence has been changed according to Reviewer suggestion.

Reviewer 2 Report

The Authors took an effort to perform a human study, what is a merit but I think much more can be obtained from the data they gathered. I suggest major revision.

  1. How the Authors can explain that the walking speed was lower in the group after 6 weeks of citrulline supplementation vs placebo group (Table 3)? Supplementation with nitric oxide donors usually boosts exercise capacity (citrulline is utilised for NO synthesis in the body)
  2. May be, sex-dependent analysis of the data in PL and CM groups after 6-week intervention would make the tendencies in Table 3 (strength, endurance) and Table 4 (haemoglobin) became statistically significant. The reviewer encourages the Authors to try analyse the data after the 6-week intervention for males and females separately.
  3. mean +/-SD or median +/- IQR should be shown were appropriate in the graphs instead mean +/- SEM. As the Authors did not see many statistically significant differences between the investigated groups in the parameters they measured, the distribution of the data should be rather shown as the dot plot rather than solid bar what would allow the reader to see the distribution of the results in the groups, were there any outsiders (did the Authors analysed the data for the presence of the outsiders?), were there any visible differences in the distribution for males and females; since the Authors investigated both males and females, it would be very interesting to see the distribution of the investigated parameters in males (i.e. black dots) and females (i.e. grey dots) in the graphs and such graphs would be much more informative.
  4. Were males (18) and females (26) equally distributed in the groups (random 9 males and 13 females in PL and the same in the CM group). If so, the numbers of male and female participants in the groups would allow to see if there were any sex-related differences to the intervention. It is highly probable in my opinion that there were ((Table 3 (strength, endurance) and Table 4 (haemoglobin, haematocrit)), but analysing the data without considering the sex covered the differences and that is why for most parameters there is no statistical significance.

Author Response

Reviewer 2 (round 1)

The Authors took an effort to perform a human study, what is a merit but I think much more can be obtained from the data they gathered. I suggest major revision.

  1. How the Authors can explain that the walking speed was lower in the group after 6 weeks of citrulline supplementation vs placebo group (Table 3)? Supplementation with nitric oxide donors usually boosts exercise capacity (citrulline is utilised for NO synthesis in the body).

ANSWER: These studies have been performed in young people. As we indicated in paragraph 4 of Discussion, our interpretation is a decreased NO production and an increase in arginase activity with age (Reference 40).

  1. May be, sex-dependent analysis of the data in PL and CM groups after 6-week intervention would make the tendencies in Table 3 (strength, endurance) and Table 4 (haemoglobin) became statistically significant. The reviewer encourages the Authors to try analyse the data after the 6-week intervention for males and females separately.

ANSWER: Unfortunately, males and females are not equally distributed in both groups. In fact, we have only one male in the placebo group, rolling out any analysis regarding sex-related differences.

  1. Mean +/-SD or median +/- IQR should be shown were appropriate in the graphs instead mean +/- SEM. As the Authors did not see many statistically significant differences between the investigated groups in the parameters they measured, the distribution of the data should be rather shown as the dot plot rather than solid bar what would allow the reader to see the distribution of the results in the groups, were there any outsiders (did the Authors analysed the data for the presence of the outsiders?), were there any visible differences in the distribution for males and females; since the Authors investigated both males and females, it would be very interesting to see the distribution of the investigated parameters in males (i.e. black dots) and females (i.e. grey dots) in the graphs and such graphs would be much more informative.

ANSWER: We have made new figures in the form of boxplots according to Reviewer suggestion. As mentioned before, the placebo group does not contain a balanced proportion of males and females. We indicate this observation as a limitation of the study (Section 2.1 of Materials and Methods).

  1. Were males (18) and females (26) equally distributed in the groups (random 9 males and 13 females in PL and the same in the CM group). If so, the numbers of male and female participants in the groups would allow to see if there were any sex-related differences to the intervention. It is highly probable in my opinion that there were ((Table 3 (strength, endurance) and Table 4 (haemoglobin, haematocrit)), but analysing the data without considering the sex covered the differences and that is why for most parameters there is no statistical significance.

ANSWER: As indicated before, males and females are not equally distributed.

Round 2

Reviewer 1 Report

The manuscript have been significantly improved . In particular , the separation of the figures into individual graphs allows a better reading of the results with a more appropriate scale, while the boxplot representation  gives a more detailed vision of the values distribution, helping to  better appreciate a positive trend with the use of citrulline supplementation combined to exercise.

Author Response

Thank you for your comments and appreciations